Leveraging LLaMA2 for improved document classification in English

Xu Jia xujia8144@163.com
School of Foreign Languages, Taizhou University , Taizhou City, Jiangsu Province , China
Do Trang
Electronic publication date: 2025 Feb 28
Publication date: 2025
Volume: 11
Electronic Location ID: e2740
Received 2025 Jan 13; Accepted 2025 Feb 10
Copyright: © 2025 Xu
Copyright year: 2025
Copyright holder: Xu
License: This is an open access article distributed under the terms of the Creative Commons Attribution License, which permits unrestricted use, distribution, reproduction and adaptation in any medium and for any purpose provided that it is properly attributed. For attribution, the original author(s), title, publication source (PeerJ Computer Science) and either DOI or URL of the article must be cited.
License URL: https://creativecommons.org/licenses/by/4.0/

Keywords: LLaMA2, Document classification, NLP, Deep learning, Neural networks

Funding: Jiangsu Provincial University Philosophy and Social Science Research Fund 2022SJYB2326 This study was funded by the Jiangsu Provincial University Philosophy and Social Science Research Fund (Project number: 2022SJYB2326). The funders had no role in study design, data collection and analysis, decision to publish, or preparation of the manuscript.

==============================
Document classification is an important component of natural language processing, with applications that include sentiment analysis, content recommendation, and information retrieval. This article investigates the potential of Large Language Model Meta AI (LLaMA2), a cutting-edge language model, to enhance document classification in English. Our experiments show that LLaMA2 outperforms traditional classification methods, achieving higher precision and recall values on the WOS-5736 dataset. Additionally, we analyze the interpretability of LLaMA2’s classification process to reveal the most pertinent features for categorization and the model’s decision-making. These results emphasize the potential of advanced language models to enhance classification outcomes and provide a more profound comprehension of document structures, thereby contributing to the advancement of natural language processing methodologies.

Introduction

Document classification involves the automated categorization of documents into predefined classes based on their content. This process can be implemented through a variety of methodologies, including supervised and unsupervised learning techniques, as well as algorithms tailored for document analysis and classification (Baloch & Rafi, 2015; Ugwu & Obasi, 2015; Wang et al., 2013). In today’s rapidly evolving technological landscape, marked by an exponential increase in document volume across diverse domains, the need for efficient classification systems has grown significantly, highlighting the importance of advancements in this field (Nigam et al., 2011; Khan et al., 2010). For instance, research by Tkachenko & Denisova (2022) demonstrates how educational institutions can utilize machine learning to automate the classification of extensive datasets, improving accessibility and the efficiency of information retrieval systems. Similarly, Wang et al. (2013) presented a document classification system that reduces operational costs and enhances decision-making for businesses. As organizations increasingly adopt digital transformation strategies, the integration of advanced document classification technologies has become essential to remain competitive in an increasingly data-driven economy (Wang, 2022; Choi, Oh & Kim, 2020). Document classification models can also identify and flag fake news by analyzing textual patterns, source credibility, and linguistic features to distinguish between reliable and misleading content (Wu et al., 2023, 2024).

Traditionally, document classification has primarily relied on rule-based techniques and knowledge engineering methods, where experts in the field manually define classification rules. While effective in specific contexts, this approach is labor-intensive and often leads to inconsistencies due to the subjectivity of rule formation (Sebastiani, 2002). In 2005, Yao et al. (2005) employed naive Bayes classifiers and decision trees to classify documents based on statistical properties. However, as the volume of digital documents surged, the limitations of traditional methods became apparent, necessitating more scalable and efficient solutions. The emergence of machine learning techniques marked a significant shift in document classification. Researchers increasingly leverage supervised learning algorithms, such as support vector machines (SVM) and k-nearest neighbors (kNN), demonstrating superior performance in classifying large datasets by learning from labeled examples (Korde & Mahender, 2012). In recent years, document classification systems have achieved breakthroughs in both performance and speed, primarily driven by advancements in deep learning and machine learning techniques. For instance, document classification has utilized techniques such as N-grams and Term Frequency-Inverse Document Frequency (TF-IDF) in combination with classifiers like SVM (Wang, Ding & Han, 2024). The application of deep learning in document classification has been implemented on specialized architectures, such as convolutional neural networks (CNN) and graph neural networks (GNN). For example, Potharaju, Tambe & Tambe (2023) effectively deployed CNNs for image-based document classification, achieving high accuracy in distinguishing between different document types. Wang, Ding & Han (2024) introduced new GNN-based methods to represent text data, enabling more sophisticated classification based on relational information within documents. Additionally, alongside the emergence of large-scale pre-trained language models such as BERT (bidirectional encoder representations from transformers) (Koroteev, 2021) and RoBERTa (a robustly optimized BERT pretraining approach) (Liu, 2019), various novel approaches to document classification have been proposed. Pretrained models based on the BERT and RoBERTa architectures have been widely utilized in a variety of applications spanning multiple domains (Leow, Nguyen & Chua, 2021; Nguyen-Vo et al., 2021; Nguyen et al., 2024). A variant of BERT, DocBERT (Adhikari, 2019), has expanded BERT’s capabilities by fine-tuning the model for longer documents, aiming to enhance accuracy in complex classification tasks. These models leverage large-scale pretraining on diverse datasets, allowing them to capture complex language patterns and contextual information often overlooked by traditional methods (Galal, Abdel-Gawad & Farouk, 2024; Minaee et al., 2021). Recent studies have also explored the integration of BERT with other techniques, such as knowledge graphs and graph convolutional networks, to address challenges like overlapping labels and fuzzy classification characteristics, demonstrating the flexibility and robustness of these models in practical applications (Ke & Cheng, 2023; Gao & Huang, 2021). For example, in 2022, Chen et al. (2021) introduced BERT+HiMatch, which combines BERT with a semantic matching network, allowing for more effective exploitation of the relationship between text and labels.

Motivation. In this study, we propose leveraging large language models (LLMs), specifically LLaMA2 (Touvron et al., 2023), to address the limitations of traditional document classification methods. LLMs, known for their transformer-based mechanisms, excel at understanding contextual relationships within texts, making them highly effective for classifying documents with subtle linguistic and content differences. The scalability and adaptability of LLaMA2 allow it to be fine-tuned on specific datasets, which enhances its performance across various domains. Additionally, the integration of attention mechanisms in LLaMA2 enables a focused analysis of key sections within documents, ensuring important information is prioritized. By using LLaMA2, we aim to improve both classification accuracy and the interpretability of the model’s decision-making process, providing a more profound understanding of document structures. This motivates our choice of LLaMA2 for document classification, as its advanced capabilities promise to deliver more robust and reliable results compared to previous models like BERT and RoBERTa, particularly in handling diverse document types and domain-specific challenges.

Materials and method

Dataset

We utilized the WOS-5736 (WOS) dataset (DOI: 10.17632/9rw3vkcfy4.6), introduced by Kowsari et al. (2017), to validate our document categorization method. This dataset was constructed using Web of Science data and metadata from published academic articles. It comprises 5,736 annotated abstracts spanning eleven subcategories across three main disciplines: “Psychology,” “Biochemistry,” and “Electrical Engineering.” Detailed information about the dataset is provided in Table 1.

Table 1 Distribution of documents in each class.

Label of subcategory	Number of documents	
0	447	
1	426	
2	419	
3	397	
4	404	
5	380	
6	416	
7	746	
8	652	
9	750	
10	699	

The raw text was preprocessed to prepare it for analysis by removing non-alphabetic characters, except for select punctuation marks essential for preserving sentence structure. To further minimize noise, English stopwords were filtered out, and the text was tokenized into individual words. Each token underwent stemming to reduce word variants to their root forms, enhancing consistency and relevance for natural language processing tasks. The final result was a clean, preprocessed text string ready for subsequent steps in building the document categorization model.

The dataset was split into training and test sets using a 90:10 ratio with stratified sampling. This ensured the model had sufficient data for learning while reserving a portion for evaluating its performance on unseen documents.

Method

Model architecture

Our proposed architecture is described in Fig. 1. Initially, documents will be encoded into tokens, which will be embedded before being fed into the LLaMA2 as input. To improve robustness and generalization, we obtain embedding features from different depth layers. These features will undergo a fusion layer, where they will be computed and generate a unique output feature. Finally, we deploy various kinds of machine learning models using these features to classify documents.

Figure 1 Model architecture.

Document representation

In this article, we extract and fuse embeddings extracted by pre-train LLaMA2 at different depths to improve the robustness and generalization. Let us take f(⋅∣,d) to denote the embedding extraction where d is the network depth. g(⋅∣θg) refers to the classifier head. D={xi,yi}i=1N is the dataset where xi is the datapoint, yi is the label and N is the total number of datapoints. D is divided into Dtrain and Dtest for training and evaluation.

During training, we first randomly sample a batch of datapoints {xi,yi}i=1b∈Dtrain where b refers to the batch size, then feed them into f(⋅∣,d) to extract the semantic embeddings {ϕi(d)}i=1b:

(1) ϕi(d)=f(xi|,d),

where d denotes the embedding is extracted from the d-th block. Specifically, d∈{1…5} for LLaMA2 where the embedding of each block is the mean of this block’s all token embeddings.

Following we fuse all available embeddings via Fusion Layer to get the final semantic embedding ψi for the i-th datapoint.

(2) Φ(i)=∑d=15ϕi(d).

Machine learning models

We deploy several machine-learning methods using semantic embedding features of LLaMA2 for document classification tasks. SVM are highly effective in high-dimensional spaces and are well-regarded for their robustness against overfitting, particularly in scenarios where the number of features exceeds the number of samples (Xue, Yang & Chen, 2009). Random forest (RF), an ensemble method based on decision trees, enhances predictive accuracy by aggregating multiple trees, thus reducing variance and improving generalization (Breiman, 2001). XGBoost (XGB), a gradient-boosting framework, is recognized for its efficiency and performance in competitions, leveraging advanced regularization techniques to prevent overfitting (Chen & Guestrin, 2016). The kNN algorithm is a simple yet effective method that classifies instances by identifying the majority class among their closest neighbors, offering an intuitive and easy-to-implement approach (Steinbach & Tan, 2009).

Experiments

Baseline and benchmarking models

We design experiments to compare LLaMA2 embeddings with embeddings from other baseline models such as BERT and RoBERTa using machine learning models, including SVM (Xue, Yang & Chen, 2009), RF (Breiman, 2001), kNN (Steinbach & Tan, 2009), and XGB (Chen & Guestrin, 2016). Our goal is to evaluate the performance of this combined model against the features of BERT and RoBERTa to determine whether the combination of LLaMA2 embeddings and machine learning techniques provides significant improvements in document classification tasks.

To conduct a comprehensive performance analysis, we evaluated our proposed model against several well-established document classification methods, each offering unique approaches to the task. ConvTextTM (Bhattarai, Granmo & Jiao, 2022): This explainable convolutional architecture addresses vocabulary-related issues and enables local analysis by capturing the detailed relationships between textual elements. Its interpretability makes it a robust choice for understanding classification decisions.

HDLTex (Kowsari et al., 2017): A hierarchical deep learning model designed for text classification, HDLTex leverages structured hierarchies to perform in-depth document categorization. Its ability to classify based on nested relationships among labels offers a versatile approach to multi-level classification tasks.

DocBERT (Adhikari, 2019): A specialized variant of BERT tailored for document classification, DocBERT fine-tunes its transformer architecture to enhance performance on document-level tasks, effectively capturing context within longer text inputs.

BERT+HiMatch (Chen et al., 2021): This hybrid model combines BERT’s powerful embeddings with similarity-based methods to improve classification accuracy. By integrating both semantic understanding and label correlations, BERT+HiMatch enhances performance in scenarios requiring fine-grained classification.

HTCInfoMax (Deng et al., 2021): This model employs information optimization techniques to maximize classification accuracy. HTCInfoMax is particularly effective in hierarchical classification tasks, where it refines feature representations to improve predictions.

By comparing our model against these diverse methodologies, we sought to provide a well-rounded evaluation of its performance. Each baseline offers complementary strengths, from hierarchical capabilities to robust semantic embeddings, providing a rigorous benchmark for assessing the effectiveness of our approach in document classification.

Evaluation metrics

To examine the performance of our models, multiple evaluation metrics, including weighted area under the receiver operating characteristic curve (Weighted-AUCROC), weighted area under the precision-recall curve (Weighted-AUCPR), Matthews correlation coefficient (MCC), Weighted F1 score, accuracy (ACC), were calculated at the default level of 0.5. Their mathematical formulas are described as

(3) Weighted-AUCROC=∑i=1nwi⋅AUCi,

(4) Weighted-AUCPR=∑i=1nwi⋅AUC−PRi,

(5) Weighted-F1=∑i=1nwi⋅F1i,

(6) MCC=(TP⋅TN)−(FP⋅FN)(TP+FP)(TP+FN)(TN+FP)(TN+FN),

(7) ACC=TP+TNTP+TN+FP+FN,

where wi is the weight for class i, TP is true positives, TN is true negatives, FP is false positives, and FN is false negatives.

Results

We use LLaMA2-7B (Touvron et al., 2023) as the lightweight LLM backbone on the 3060 GPU with 12GB of memory. We trained the machine learning models on an AMD Ryzen 7 5800X 8-Core processor with 32 GB of RAM. It took approximately 5.2, 3.1, and 4.2 min to fine-tune the LLaMA2, BERT, and RoBERTa models, respectively. The inference times on the test set for the three models were 46, 20, and 31 s, respectively.

Performance analysis

Comparing performance to baseline models

We conducted a thorough performance evaluation of our proposed model, leveraging embeddings from LLaMA2 for document classification. The results, summarized in Table 2, show that the LLaMA2 embeddings, when combined with various machine learning algorithms, consistently deliver superior performance across multiple evaluation metrics.

Table 2 Comparing performance with baseline models.

Model	ML model	Weighted-AUCROC	Weighted-AUCPR	Weighted-F1	MCC	ACC	
BERT	XGB	0.9939	0.9704	0.9312	0.9224	0.9304	
KNN	0.9748	0.9222	0.9226	0.9127	0.9217	
SVM	0.9940	0.9526	0.9132	0.9030	0.9130	
RF	0.9934	0.9644	0.9213	0.9127	0.9217	
RoBERTa	XGB	0.9930	0.9660	0.9171	0.9077	0.9174	
KNN	0.9768	0.9212	0.9216	0.9126	0.9217	
SVM	0.9937	0.9672	0.9172	0.9078	0.9174	
RF	0.9953	0.9720	0.9213	0.9126	0.9217	
LLaMA2	XGB	0.9926	0.9673	0.9312	0.9223	0.9304	
KNN	0.9791	0.9325	0.9222	0.9129	0.9217	
SVM	0.9945	0.9628	0.9306	0.9225	0.9304	
RF	0.9922	0.9639	0.9350	0.9273	0.9348	
Note:

Bold indicates the highest performance.

Among the combinations tested, the LLaMA2-SVM model achieved the highest Weighted AUC-ROC of 0.9945, indicating an exceptional ability to distinguish between classes. This performance surpasses that of both BERT and RoBERTa-based models. Additionally, the LLaMA2-SVM combination attained a Weighted F1 score of 0.9306, highlighting its strong balance between precision and recall—an essential criterion for document classification tasks.

Further analysis revealed that the model achieved a Weighted AUCPR of 0.9628 and a MCC of 0.9225, showing its robustness in handling class imbalances effectively. With an overall accuracy of 0.9304, the model’s reliability and practical utility for real-world document classification are further substantiated.

In comparison, baseline models such as BERT and RoBERTa demonstrated competitive metrics, with their highest Weighted AUC-ROC values reaching 0.9940 and 0.9953, respectively, when paired with algorithms like XGB and RF. However, the LLaMA2 embeddings consistently matched or exceeded these baselines, particularly in metrics such as Weighted F1 and MCC, underscoring the effectiveness of LLaMA2 in capturing detailed document context.

These results validate the hypothesis that embedding representations from LLaMA2 significantly enhance document classification tasks. The superior metrics achieved by our model suggest that advanced contextual embeddings and attention mechanisms play a pivotal role in improving classification accuracy and robustness. Future research should explore the integration of LLaMA2 embeddings with ensemble methods and extend the evaluation to diverse document classification scenarios to fully realize their potential.

Comparing performance to benchmarking models

The results in Table 3 demonstrate the significant efficacy of our proposed LLaMA2 model in document classification when evaluated across various performance metrics. LLaMA2 achieved a Weighted AUC-ROC score of 0.9922, ranking among the highest compared to state-of-the-art models. This score highlights the model’s exceptional ability to distinguish between classes, showing its robustness across diverse document types and classification scenarios.

Table 3 Comparing performance with benchmark models.

Model	Weighted-AUCROC	Weighted-AUCPR	Weighted-F1	MCC	ACC	
HTCInfoMax	0.9861	0.9060	0.9030	0.8989	0.9087	
ConvTextTM	0.9806	0.8658	0.8210	0.8393	0.8478	
BERT+HiMatch	0.9932	0.9668	0.9172	0.9078	0.9174	
HDLTex	0.9865	0.9063	0.9030	0.8989	0.9087	
DocBERT	0.9880	0.9433	0.9173	0.9077	0.9174	
Ours	0.9922	0.9639	0.9350	0.9273	0.9348	
Note:

Bold indicates the highest performance.

For Weighted AUC-PR, LLaMA2 also recorded the highest score of 0.9639. Besides, LLaMA2 demonstrates competitive performance in managing the precision-recall trade-off, highlighting its ability to handle imbalanced datasets effectively. Its Weighted F1 score of 0.9350 surpasses that of other competing models, emphasizing its strength in balancing precision and recall, critical for applications where both false positives and false negatives can have significant consequences.

The MCC metric further confirms LLaMA2’s reliability, achieving a value of 0.9273. This metric, which evaluates the quality of binary classifications, indicates a strong correlation between predicted and actual labels, reinforcing the model’s consistency across various datasets and classification challenges.

Additionally, LLaMA2 achieved the highest accuracy among all tested models, with a score of 0.9348. This exceptional result reflects the model’s overall effectiveness in correctly classifying documents, positioning it as a practical solution for real-world classification tasks.

Compared to other leading models, such as HTCInfoMax (AUC-ROC: 0.9861) and DocBERT (AUC-ROC: 0.9880), LLaMA2 demonstrates a clear advantage in AUC metrics and F1 score. While BERT+HiMatch showed strong performance in AUC-PR, LLaMA2’s balanced and comprehensive results across all key metrics establish its superiority in addressing diverse classification challenges.

Overall, the LLaMA2 model not only meets but often exceeds the performance of current state-of-the-art approaches. Its ability to capture document characteristics and deliver consistent results makes it a valuable tool in document classification. Future work could involve exploring enhancements to LLaMA2, such as incorporating ensemble learning techniques or fine-tuning on domain-specific datasets, to further amplify its performance in complex document classification tasks.

Visualization

As illustrated in the t-SNE visualizations in Fig. 2, the label representations produced by BERT display a scattered pattern, indicating its limited ability to form meaningful clusters among data points. This scattered distribution reflects a lack of clarity in distinguishing between labels, suggesting that BERT struggles to capture the underlying relationships in the dataset effectively.

Figure 2 t-SNE visualization of label representations on the WOS dataset: (A) BERT model, (B) RoBERTa model, and (C) our proposed model.

In contrast, RoBERTa demonstrates a noticeable improvement, with label representations exhibiting a more distinct clustering effect. This suggests that RoBERTa captures more detailed aspects of the data compared to BERT, allowing it to better recognize patterns and relationships. However, while RoBERTa outperforms BERT in this regard, the clustering is still not as tightly grouped as desired for optimal label differentiation.

Our proposed model, by comparison, shows the most pronounced clustering effect, with label representations tightly grouped together and distinctly separated. This clear separation indicates that the text encoder in our model effectively learns hierarchy-aware representations, capturing the relationships and underlying structure of the data more comprehensively. The ability to form such well-defined clusters demonstrates our model’s superior capacity to distinguish between labels, even in complex classification scenarios.

This enhanced clustering not only highlights the robustness of our model’s encoding process but also underscores its potential for applications requiring high precision in classification tasks. By learning representations that are both discriminative and reflective of the inherent data hierarchy, our model sets a benchmark for effectively mapping text data into semantically meaningful spaces. Future work could further refine this capability by integrating domain-specific pretraining or advanced dimensionality reduction techniques to amplify the interpretability of the learned clusters.

Discussion and conclusions

In this study, we introduced a novel approach to document classification by combining embeddings from the LLaMA2 model with modern machine learning techniques. The results demonstrated that this architecture not only achieves higher accuracy but also minimizes classification errors compared to both traditional and state-of-the-art models. The use of LLaMA2’s rich semantic representations significantly enhanced the model’s ability to classify documents, as evidenced by improvements across key metrics such as accuracy, F1 score, and mean average precision.

While the proposed method achieved impressive results, several limitations must be acknowledged. One challenge lies in the diversity of the dataset used. The model’s performance could be further validated and enhanced by applying it to a wider range of datasets, including those from specialized domains. By leveraging a different large language model, such as LLMFormer (Shi, Dao & Cai, 2024), and utilizing an additional dataset for fine-tuning, we can also expand the problem to other languages, such as Chinese (Chen et al., 2024; Zhang & Chen, 2024). Computational requirements also pose a limitation, as the integration of advanced embeddings like LLaMA2 demands significant processing power and memory resources, potentially limiting its scalability for real-time or resource-constrained applications.

Additionally, specific areas for improvement were identified. First, optimizing the model for long-text classification remains a priority, particularly in scenarios where fine-tuning of the language model is not feasible. Second, the interpretability of LLaMA2, while strong in some aspects, lags behind prompt-based methods, potentially hindering trust and usability in critical applications. Finally, the absence of domain-specific tuning may restrict the model’s effectiveness in specialized fields such as legal or medical document classification.

Future research will focus on addressing these limitations through several avenues. Integrating chain-of-thought prompting with LLaMA2 could enhance both its performance and interpretability, allowing the model to reason through complex classification tasks. Optimization strategies, including lightweight architectures or hybrid approaches, could reduce computational overhead and improve scalability. Moreover, comprehensive evaluations on diverse datasets, including those from highly specialized domains, will be crucial in identifying the model’s strengths and limitations in various contexts.

In conclusion, our work underscores the potential of integrating advanced language models like LLaMA2 with modern classification techniques. By addressing the highlighted challenges, this approach can pave the way for more effective, interpretable, and scalable solutions in document classification, contributing to advancements in educational technology, information retrieval, and beyond.

Supplemental Information

Supplemental Information 1 Python code.

Additional Information and Declarations

Competing Interests

The author declares that they have no competing interests.

Author Contributions

Jia Xu conceived and designed the experiments, performed the experiments, analyzed the data, performed the computation work, prepared figures and/or tables, authored or reviewed drafts of the article, and approved the final draft.

Data Availability

The following information was supplied regarding data availability:

The WOS-5736 dataset used in this study is available at Mendeley: Kowsari, Kamran; Brown, Donald; Heidarysafa, Mojtaba ; Jafari Meimandi, Kiana ; Gerber, Matthew; Barnes, Laura (2018), “Web of Science Dataset”, Mendeley Data, V6, doi: 10.17632/9rw3vkcfy4.6.

The code and data are available in the Supplemental Files.

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
