# Peer review of "Leveraging LLaMA2 for improved document classification in English"

_PeerJ Computer Science, doi:10.7717/peerj-cs.2740_

## Round 0.1 · original submission · Major Revisions

Please revise the manuscript based on the comments from the reviewers. Consider including an extra experiment and including more relevant studies.

Reviewer 1 ·

Basic reporting

- English used is OK. The language in the paper is clear and unambiguous.
- While Intro & background show context, I believe the referenced still can be improved by including more relevant studies.
- No deviations need more clarity.
- The Introduction adequately introduces the subject and make it clear what the motivation is.
- All the related terms were clearly defined in the paper.

Experimental design

- The paper content is within Aims and Scope of the journal and article type.
- Code and data are mentioned and share; this is a very good point.
- The paper mentions sufficient information and bout how the data was preprocessed, the selection of the evaluations methods, the evaluation metrics and the model.
- However, it is better if the information about the number of samples for each class after data splitting is included in the paper.
- Also, the some timing information when training the model, especially when finetuning the LLaMA2 model, should be provided.

Validity of the findings

- The rationale & benefit to the field of the work was clearly stated in the paper.
- The conclusions are well stated, and the authors did mention some limitations of the study.
- The experiments and evaluations were performed quite satisfactorily, although it is better to add more comparisons with some of other methods.
- Somehow the conclusion identifies unresolved questions, limitations, future directions. Nevertheless, this part still can be improved.

Additional comments

- This is a good work. My recommendation is somewhere in between Minor and Major revisions. Mostly about adding more relevant contents. Although some extra experiments were suggested, this is not totally needed.

Cite this review as

Reviewer 2 ·

Basic reporting

- The manuscript is well-organized and composed in clear, formal English, demonstrating a strong command of academic writing. The literature review is thorough and offers a comprehensive overview of the topic, effectively contextualizing the research. The tables and figures are thoughtfully designed and serve as valuable tools for presenting the findings in an accessible manner. That said, certain sections of the main text could be enhanced with more detailed explanations or elaborations, as noted in the comments below.

Experimental design

- The dataset was split into training and test sets using a 90:10 ratio, but could you clarify whether this division was based on stratified sampling or purely random sampling?
- The paper would be clearer if it included a visualization or detailed description of the distribution of sentence lengths within the dataset.

Validity of the findings

- How quickly can the model perform during the inference phase?
- While the comparisons provided in the paper are fairly comprehensive, I believe the analysis could be strengthened by including additional related models. For instance, the authors might consider comparing their proposed model with BERT+HiMatch, the approach introduced by Chen et al. (2021), which is referenced in the Introduction section of the paper.

Additional comments

- Please review the capitalization in the reference list. Some terms, such as kNN, SVM, BERT, etc., should be capitalized correctly.
- While the authors have touched on the limitations and future directions of the study, I believe this section could be expanded by incorporating insights or comparisons from additional relevant papers.

Cite this review as

---

## Round 0.2 · accepted · Accept

The authors have made significant revisions to the manuscript and thoroughly addressed all reviewer comments. Based on the reviewers' suggestions, I recommend accepting this version for publication.

Reviewer 1 ·

Basic reporting

- The revised manuscript satisfies the requirements for basic reporting.
- The language used throughout is clear, unambiguous, and professional, ensuring readability and precision.
- The introduction and background provide sufficient context, effectively setting the stage for the study, while the literature is well-referenced and relevant, demonstrating a solid foundation in existing research.
- The structure adheres to PeerJ standards and discipline norms, with any deviations intentionally made to enhance clarity.
- The introduction adequately introduces the subject, clearly outlining the motivation for the study and its significance within the broader field.
Overall, the manuscript meets the expected standards for reporting and presentation.

Experimental design

The revised manuscript satisfies the requirements for experimental design:
- The content aligns well with the Aims and Scope of the journal and the chosen article type.
- All relevant code and data have been provided, ensuring transparency and reproducibility.
- The discussion on data preprocessing is included and is sufficient for the study's context, addressing necessary steps such as cleaning, normalization, or transformation where applicable.
- The evaluation methods, assessment metrics, and model selection processes are adequately described, with clear justification for their use in relation to the research objectives.

Validity of the findings

The findings are robust and well-supported by experimental results, including one extra experiment, as well as by comparisons with other methods.

Additional comments

All of my comments have been addressed. I have no further comment.

Cite this review as

Reviewer 2 ·

Basic reporting

- The manuscript is clear and well-structured. The literature review is comprehensive, and tables and figures effectively support the findings.
- All previous comments regarding adjustment have been fully addressed.

Experimental design

- The explanation of the dataset splitting method and the visualization have fixed for improving the clarity of the methodology.
- The experimental design is robust and well-supported.

Validity of the findings

- The inference speed has been clarified, and additional comparisons, strengthen the analysis.
- The findings are valid and well-presented after adjustment.

Additional comments

- References are correctly formatted, with proper capitalization. The expanded discussion on limitations and future directions adds valuable insights.

Cite this review as